# Diagnostic Accuracy of HPV Detection in Patients with Oropharyngeal Squamous Cell Carcinomas: A Systematic Review and Meta-Analysis

**DOI:** 10.3390/v13091692

**Published:** 2021-08-26

**Authors:** Kathrine Kronberg Jakobsen, Amanda-Louise Fenger Carlander, Simone Kloch Bendtsen, Martin Garset-Zamani, Charlotte Duch Lynggaard, Christian Grønhøj, Christian von Buchwald

**Affiliations:** Department of Otorhinolaryngology, Head and Neck Surgery and Audiology, Rigshospitalet, University Hospital of Copenhagen, 2100 København, Denmark; amanda.louise.fenger.carlander@regionh.dk (A.-L.F.C.); simone.kloch.bendtsen.01@regionh.dk (S.K.B.); martin.garset-zamani@regionh.dk (M.G.-Z.); charlotte.duch.lynggaard@regionh.dk (C.D.L.); christian.groenhoej@regionh.dk (C.G.); christian.von.buchwald@regionh.dk (C.v.B.)

**Keywords:** human papillomavirus, HPV, oropharyngeal cancer, diagnostic

## Abstract

The aim of the study was to evaluate the diagnostic accuracy of Human Papillomavirus (HPV) techniques in oropharyngeal cancer. PubMed, EMBASE, the Cochrane Library and clinicaltrials.org were systematically searched for studies reporting methods of HPV detection. Primary outcomes were sensitivity and specificity of HPV detection. In this case, 27 studies were included (*n* = 5488, 41.6% HPV+). In this case, 13 studies evaluated HPV detection in tumour tissue, nine studies examined HPV detection in blood samples and five studies evaluated HPV detection in oral samples. Accuracy of HPV detection in tumour tissue was high for all detection methods, with pooled sensitivity ranging from 81.1% (95% CI 71.9–87.8) to 93.1% (95% CI 87.4–96.4) and specificity ranging from 81.1% (95% CI 71.9–87.8) to 94.9% (95% CI 79.1–98.9) depending on detection methods. Overall accuracy of HPV detection in blood samples revealed a sensitivity of 81.4% (95% CI 62.9–91.9) and a specificity of 94.8% (95% CI 91.4–96.9). In oral samples pooled sensitivity and specificity were lower (77.0% (95% CI 68.8–83.6) and 74.0% (95% CI 58.0–85.4)). In conclusion, we found an overall high accuracy for HPV detection in tumour tissue regardless of the HPV detection method used. HPV detection in blood samples may provide a promising new way of HPV detection.

## 1. Introduction

The incidence of oropharyngeal squamous cell carcinomas (OPSCCs) caused by human papillomavirus (HPV) is increasing worldwide [1,2]. Previously, the main causes of OPSCCs were smoking and alcohol consumption but today up to 70% of cases in most parts of the Western world are associated with HPV-driven carcinogenesis [3,4,5,6,7]. HPV+ OPSCC has a unique epidemiologic profile, molecular composition and histopathological features compared to the tobacco and alcohol associated OPSCC [3,8,9,10]. Patients are commonly younger, with fewer co-morbidities and have a better prognosis [11,12,13]. A surrogate marker for HPV infection is tumour suppressor protein p16 positivity (p16+). p16+ OPSCC has shown better prognosis compared to p16 negative (p16−) tumours. However, double positivity, i.e., tumours being positive for both HPV and p16 have shown better prognostication compared to a single marker of positivity [14].

Several techniques to evaluate HPV positivity exist. These includes p16 evaluation by immunohistochemistry (IHC), detection of HPV DNA by in situ hybridisation (ISH) or by polymerase chain reaction (PCR), E6/E7 HPV mRNA evaluation by ISH and reverse transcriptase-PCR (RT-PCR), or a combination of the above-mentioned methods. E6/E7 HPV mRNA evaluation is considered the golden standard to assess HPV positivity, as this technique detects oncogenic transcriptional active HPVs, but the test is expensive and technically challenging to perform [15]. On the other hand, p16 assessment is the most used technique in clinical settings as it is easy to conduct and to interpret, is less expensive and widely available [15,16]. This has led to p16 being included in the 8th edition of American Joint Committee on Cancer (AJCC) and the Union for International Cancer Control (UICC) tumour, node, metastasis (TNM) staging system of OPSCC, where p16+ tumours now have a novel staging system distinct from the staging of p16− tumours [17]. The recommendation from The American Society of Oncology (ASCO) for defining a tumour as p16+ is by a cut-off of 70% nuclear and cytoplasmic staining [15]. However, several studies have shown disparities in the cut-off level defining a tumour as p16 positivity [15,18].

The definition of HPV+ OPSCC is a critical issue as treatment de-escalation in patients with HPV+ tumours is currently being investigated in clinical trials to avoid unneeded treatment-related side effects, overtreatment, and to minimize the risk of treatment-related acute and long-term morbidity in this patient group. However, this should be performed without misallocating patients with less favourable prognosis to less treatment.

In addition, several new techniques for assessing HPV positivity without the need of an invasive biopsy of tumour tissue, e.g., by liquid biopsy using saliva or blood are advancing which would be a readily available way of detecting HPV. Circulating tumur DNA (ctDNA) from virus-induced cancers has previously been shown to be clinically useful as a diagnostic test for oncovirus-driven cancers, such as Hepatitis B virus (HBV)-induced hepatocellular carcinoma [17] and Epstein-Barr virus (EBV)-induced nasopharyngeal carcinoma (NPC) [18,19]. HPV DNA has also been shown to be present in plasma in patients with HPV-induced cervical cancer but absent in patients with cervical dysplasia [20,21].

Evolution in laboratory techniques is rapidly evolving, experience with p16 detection is increasing and new detection standards are continuously being presented. An update on the recent knowledge in HPV detection is a timely needed study. Furthermore, a comparison of the diagnostic accuracy in different specimens and a ranking of these are warranted.

The aim of this study was to systematic review the literature on methods of HPV detection and to assess the diagnostic accuracy for HPV detection in patients with OPSCC based on detection methods and in different sample types.

## 2. Materials and Methods

This systematic review and meta-analysis was conducted with reference to the Preferred Reporting Items for Systematic Reviews and Meta-Analyses (PRISMA) statement [19].

One author (KKJ) systematically searched the PubMed, EMBASE, Cochrane databases and clinical trials.org for articles in English and Scandinavian language. The search was last updated on the 28 May 2021. We included original studies comprising OPSCC patients investigating diagnostic methods of HPV detection published within the last five years. Studies comprising patients with OPSCC along with other head and neck cancer subsites were included, if they provided information specifying the results of the diagnostic accuracy of HPV detection for the OPSCC patients. Studies were excluded if they included less than 10 OPSCC cases and if the HPV detection method, including the definition of HPV positivity and p16 positivity, was not defined.

The search term was phrased broadly to identify relevant references. The following keywords were used to build the search: (Oropharyn* cancer or oropharyn* neoplasm or oropharyn* carcinoma or oropharyn* malignancy or oropharyn* tumour or oropharyn* tumour) AND (HPV or human papillomavirus or human papilloma virus or p16 or papillomavirus or p16 or cdkn2a or cyclin-dependent kinase inhibitor p16 or p16 genes) AND (diagnosis or diagnostic)). The search strategy in PubMed included MeSH terms.

We collected information on study type, diagnostic methods, reference methods, sample type, HPV type and the sensitivity and specificity of the diagnostic methods.

Statistical analyses were performed using R studio, version 1.2.5. We generated paired forest plots depicting sensitivity and specificity estimates across studies. We conducted a meta-analysis using the bivariate model in R studio by using the mada package function reitsma [20]. The model is a linear mixed model with known variance of the random effects. In the bivariate model, the logit transformed sensitivities and specificities and the correlation are modeled directly. The model accounts for sampling variability within studies and also account for between-study variability through the inclusion of random effects. The bivariate approach incorporates any correlation that might exist between two measures using a random effects approach.

## 3. Results

The literature search generated 1513 articles, of which 24 were enrolled. Three additional articles were identified through reference lists (Figure 1).

A total of 1389 articles were excluded based on screening of the title and abstract. Of these, we excluded studies that did not focus on the diagnostic accuracy of HPV detection (*n* = 975), other reviews, case reports and editorials (*n* = 218) and lastly, studies regarding HPV diagnostics in other patient groups than OPSCC (*n* = 196). Thus, 119 articles were assessed by full-text. In this case, 92 studies were not included as they did not concern diagnostic methods (*n* = 50), did not report diagnostic accuracy (15), did not specify results for OPSCC patients (*n* = 11), did not define HPV or p16 positivity (*n* = 9) or included less than 10 patients (*n* = 7).

Finally, 27 studies were included comprising a total of 5488 patients diagnosed with OPSCC (41.6% HPV+). Three studies included a non-OPSCC control group (*n* = 229) consisting of head and neck cancer patients with a cancer located at another subsite than the oropharynx (*n* = 74), healthy controls (*n* = 75), patients with Warthin’s tumour (*n* = 20) or branchial cleft cyst (*n* = 10). The studies including non-OPSCC patients were excluded in the meta-analysis. In this case, 14 studies were European, 10 were US and Canada based and three studies were Asian (Table 1).

### 3.1. Diagnostic Accuracy of HPV in Tissue Samples

Nine studies evaluated HPV testing in formalin-fixed paraffin-embedded (FFPE) tissue [21,22,23,24,25,26,27,28,29], and five studies evaluated HPV detection in fine needle aspiration (FNA) [30,31,32,33,34]. Reference methods varied between studies; seven studies used p16 IHC as reference [24,25,27,30,33,34], three studies used HPV RNA as reference, where one study detected HPV RNA by ISH [22] and the other two studies detected HPV RNA by PCR [23,26]. Another three studies used HPV DNA by PCR [28,29,32] as reference and finally, two studies combined two detection methods for reference; p16 IHC combined with HPV DNA PCR [21] and p16 IHC combined with HPV DNA PCR and/or HPV E6 seropositivity [31], respectively (Table 1).

Of the nine studies evaluating HPV detection in FFPE, six studies evaluated more than one detection method [22,24,25,26,28,29]. Five studies investigated accuracy of HPV RNA ISH, four studies investigated accuracy of p16 IHC, five studies investigated accuracy of HPV DNA PCR and four studies investigated the accuracy of HPV DNA ISH. Sensitivity was overall high and ranged from 74% (95% CI 64–82%) [25] to 99% (95% CI 89–100) [21] (Figure 2).

Specificity ranged considerably, with the lowest specificity reported as 55% (95% CI 38–71) [29], and three studies reported specificity as high as 100% [25,26,27]. Stratified by detection method, the meta-analysis revealed that the pooled sensitivity was highest for HPV RNA ISH, 93.1 (95% CI 87.4–96.4) and lowest for HPV DNA ISH, 81.1% (95% CI 71.9–87.8). Regarding the pooled specificity the method with the highest specificity was the HPV DNA ISH detection method (94.9% (95% CI 79.1–98.9)) and the lowest was found for HPV DNA PCR (81.1 (95% CI 71.9–87.8)). The reference detection method varied between studies (Table 2).

The five studies evaluating the diagnostic accuracy in FNA comprised few patients (*n* = 195) with the majority of patients being HPV+ OPSCC (85.6%). Four of the five included studies reported a specificity of 100% [30,31,32,33] calculated on the basis of a total of 15 patients with HPV- OPSCC. One study did not include HPV- OPSCC patients and specificity was calculated on the basis of seven patients with oral squamous cell carcinoma (OSCC), 20 Warthin´s tumours and 20 branchial cleft cysts [32]. The studies all reported sensitivity above 94% (Figure 3). The five studies evaluating diagnostic accuracy in FNA were not included in the meta-analysis, as the study numbers were too few to conduct a valid meta-analysis.

### 3.2. Diagnostic Accuracy of Detecting HPV in Blood Samples

Nine studies (*n* = 1353, 77.6% HPV+) examined the diagnostic accuracy of HPV detection by liquid biopsy using blood samples. One study collected the blood samples both at time of diagnosis and during treatment and evaluated the accuracy of the test regardless of the time of blood collection [35], the other eight studies collected blood samples before treatment initiation [36,37,38,39,40,41,42,43]. Five studies tested HPV in plasma [36,37,38,39,40], two studies reported detection in blood [35,41] and two studies investigated HPV in serum [42,43].

Studies predominantly evaluated detection of circulating HPV DNA in the blood. However, one study examined the accuracy of detecting HPV16 E6/E7 expression in circulating tumour cells (CtCs) [41]. The study addressing diagnostic accuracy of HPV expression in CtCs had a significantly lower sensitivity compared to the remaining studies (Figure 4).

In the pooled analysis of sensitivity and specificity, the overall sensitivity was 81.4% (95% CI 62.9–91.9), and the overall specificity was 94.8% (95% CI 91.4–96.9) covering all studies regardless of detection method and reference [35,36,38,40,41,42,43] (excluding the two studies with non-OPSCC patients [37,39]).

### 3.3. Diagnostic Accuracy of Detecting HPV in Oral Samples

Five studies evaluated the diagnostic accuracy of HPV detection in oral samples (Table 1) corresponding to a total of 543 patients with OPSCC. Three studies collected oral samples by oral rinse [44,45,46], one study used cytologic brush of the tumour area [47] and one study combined saliva collection with oral swabs [34]. Three studies used p16 IHC combined with HPV DNA as reference [44,46,47], one study only used p16 IHC as reference [34] and one study used mRNA E6 and E7 as reference [45].

A considerable variability in sensitivity and specificity estimates were observed among the studies ranging from 60.9% (95% CI 40.8–77.8) [47] to 86.1% (95% CI 80.4–90.3) [45] in sensitivity, and from 50% (95% CI 27–73) [46] to 91% (95% CI 62–98) [34] in specificity (Figure 5).

Regardless of oral sample collection method and reference, method the meta-analysis revealed a pooled sensitivity and specificity of 77.0% (95% CI 68.8–83.6) and 74.0% (95% CI 58.0–85.4), respectively. When restricted to the four studies using p16 IHC as a reference, the pooled sensitivity and specificity was 77.6% (95% CI 67.8–78.7) and 72.1% (95% CI 49.1–87.4).

## 4. Discussion

This systematic review and meta-analysis evaluated the diagnostic accuracy of HPV detection in patients with OPSCC. As laboratory techniques are evolving rapidly and new detection methods continuously are being introduced, this is an area in need of an update in the current literature. A ranking and comparison of the diagnostic accuracy in different specimens are furthermore needed.

As p16-status is an important factor in staging OPSCC [17], and as clinical trials on treatment de-escalation for HPV+ OPSCC are continuously being introduced, precise detection methods of HPV is of immense importance.

We included a total of 27 studies with varying specimens, methods of HPV detection and references for the latter. We first looked at studies evaluating the diagnostic accuracy of HPV detection in tumour tissue. Two studies evaluated the diagnostic accuracy of combining two detection methods, i.e., p16 IHC combined with HPV DNA PCR. Both studies reported high sensitivity of 93% (95% CI 74–98%) and 86% (95% CI 76–92%), respectively. A similar systematic review [48] also found that combination of diagnostic tests represented the most attractive testing strategy in HPV-related OPSCC. However, it should be noted that only two of the included studies on diagnostic accuracy in FFPE used combined detection methods. We did also find a high diagnostic accuracy in studies where only one diagnostic test was used.

Of importance, we excluded nine studies where the definition of p16 positivity was not specified. The exclusion may have had an impact on the results of our review. It is incredibly important to specify the p16 positivity, since it has been shown that to achieve the highest correlation between p16 and HPV results, a staining of >70% of tumour cells to classify the tumour as p16 positive is advised [49]. The enrolled studies evaluating p16 in tumour cells used a limit >70% staining, except two studies using a cut-off value of 66% and 50%.

Recently, the ability to detect HPV in liquid biopsies was introduced as a novel, non-invasive method of HPV detection. The use of a liquid biopsy for cancer detection has shown encouraging results in both colorectal cancer and bladder cancer [50,51,52]. In contrast to HPV-related cervical cancer, precancerous lesions are lacking in OPSCC and reliable screening methods are thus needed. At present, only a few studies on the use of liquid biopsies in OPSCC exists. Our review indicates that the diagnostic accuracy of HPV detection in blood samples constitutes a promising tool in HPV detection with an overall sensitivity of 81.4% (95% CI 62.9–91.9) and an overall specificity of 94.8% (95% CI 91.4–96.9). Methods used for estimating HPV positivity in primary OPSCC patients varied which could partly explain some of the heterogeneity in sensitivity and specificity. The conclusion that HPV detection in liquid biopsy obtained from OPSCC patients may have a promising role correlates well with a closely related meta-analysis [53]. It is worth noticing that despite of the high sensitivity and specificity, the low prevalence of OPSCC in the general population will result in a low positive predictive value leading to a low current value of HPV as a population-wide cancer screening biomarker as described by the International Agency for Research on Cancer (IARC) and the US National Cancer Institute (NCI) [54].

When looking at the diagnostic accuracy in oral samples obtained from OPSCC patients, our study revealed a lower diagnostic accuracy than the other specimen types with a sensitivity and specificity of 77.6% (95% CI 67.8–78.7) and 72.1% (95% CI 49.1–87.4), respectively. Variability amongst the studies detecting HPV in oral samples varied considerably. A similar meta-analysis [55] investigating the diagnostic accuracy of HPV detection in oral samples from OPSCC patients found a lower sensitivity 55% (95% CI 25–82%), but with a higher specificity 94% (95% CI 85–98%). The difference could be explained by the fact that their study differed from ours as they enrolled non-OPSCC head and neck cancer patients in their cohort. The International Agency for Research on Cancer (IARC) and the US National Cancer Institute (NCI) reported similar findings with ranging sensitivity and specificity [54].

In general, the included studies varied in regards to reference method as well as method of detecting HPV, which comprises a significant limitation when comparing the diagnostic accuracy between studies. This might be one possible explanation for the variation in accuracy across the included studies. The pooled sensitivity and specificity should thus be interpreted with caution as accuracy can vary depending on testing method and reference method. To investigate the accuracy of the specific detection methods in order to circumvent the variability and uncertainty different detection methods might bring to the meta-analyses, we performed a sub-analysis of the diagnostic accuracy stratified on detection method for the studies assessing accuracy in FFPE. We found that sensitivity and specificity in general were high for all detection methods with sensitivity ranging from 81.1 (95% CI 71.9–87.8) to 93.1 (95% CI 87.4–96.4), and specificity ranging from 81.1 (95% CI 71.9–87.8) to 94.9 (95% CI 79.1–98.9). It was not possible to conduct sub-analysis for accuracy in liquid biopsy and FNA due to the lower study numbers. We did not account for the different reference methods in the meta-analysis, as it would have resulted in too few studies to perform a valid meta-analysis. In general, grouping studies regardless of the reference methods when comparing sensitivity and specificity of diagnostic testing in meta-analyses [53,55] is a limitation and an ongoing challenge. It is difficult to circumvent, as further stratification according to reference method would lead to very few studies resulting in new limitations and limiting the number of eligible studies so considerable that a meta-analysis could not be performed. Further studies on diagnostic accuracy of HPV detection with similar reference methods are thus warranted.

We included studies published within the last five years to avoid excessive variation in the detection methods between studies as p16 positivity previously ranged considerably and a large part of studies used a minimum of 5–69% staining [49] before ASCO published guidelines for defining a tumours as p16+ by a cut-off of 70% nuclear and cytoplasmic staining [15]. This is however also a limitation to the study that should be noted.

## 5. Conclusions

In conclusion, our systematic review evaluating HPV detection methods in patients with OPSCC showed an overall high sensitivity and specificity of HPV detection in FFPE for both RNA ISH, DNA ISH, DNA PCR and p16 IHC. HPV detection by liquid biopsy and blood samples provides a promising, less invasive method of HPV detection and both sensitivity and specificity were high, thus highlighting HPV detection in blood samples as a promising novel tool of HPV detection. HPV detection in blood samples showed an overall sensitivity of 81.4% (95% CI 62.9–91.9), and an overall specificity of 94.8% (95% CI 91.4–96.9) which is thus comparable to the sensitivity and specificity of HPV detection in FFPE where the sensitivity was ranging from 81.1% (95% CI 71.9–87.8) to 93.1 (95% CI 87.4–96.4) and the specificity was ranging from 81.1 (95% CI 71.9–87.8) to 94.9% (95% CI 79.1–98.9.

Lastly, results on the accuracy of HPV detection in FNA and in oral samples were scarce and varied considerably, and evidence on the use of oral samples in HPV detection is currently not substantial enough to highlight it as an acceptable diagnostic tool. In summary, larger studies with homogenous study designs are required to further explore the diagnostic applicability of various HPV detection methods in patients with HPV+ OPSCC.

## Figures and Tables

**Figure 1 viruses-13-01692-f001:**
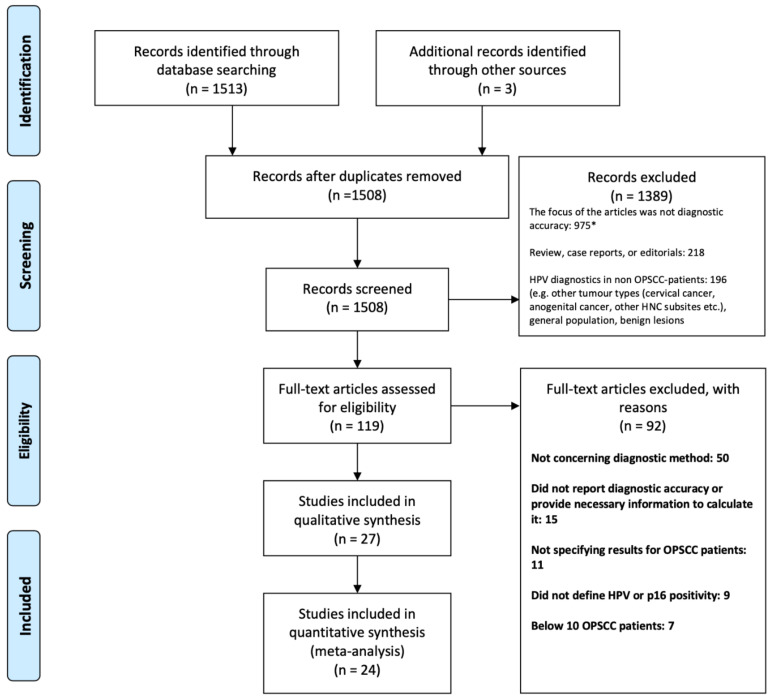
Prisma flow diagram. Abbreviation: Oropharyngeal squamous cell carcinoma (OPSCC). * Studies focused on the epidemiological aspects of OPSCC (e.g., survival and incidence in patients with HPV+ and HPV- OPSCC) (*n* = 385), treatment of HPV+ OPSCC (*n* = 230), biomarkers, molecular characterization and gene expression in HPV+ tumours (*n* = 218), imaging of OPSCC tumours (*n* = 53), quality of life in OPSCC patients (*n* = 26), HPV vaccines (*n* = 14), validation of assays for HPV detection (*n* = 11), costs of HPV detection (*n* = 9), concerning oral health (*n* = 4), others (*n* = 25), e.g., HPV clearance profile, HPV load in relation to tumour size and HPV genotypes.

**Figure 2 viruses-13-01692-f002:**
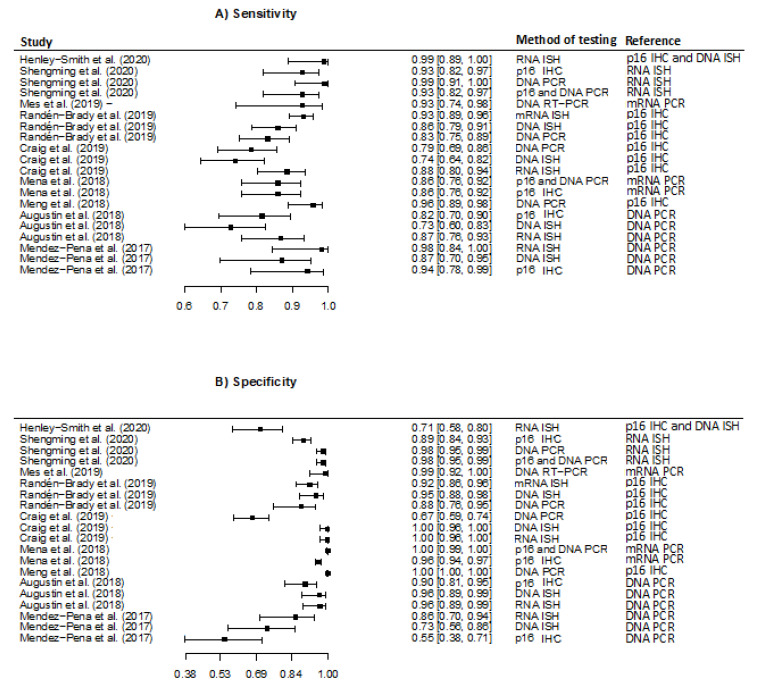
Forest plots of the meta-analysis estimating (**A**) Sensitivity with 95% confidence intervals, (**B**) Specificity with 95% confidence intervals in studies evaluating detection of Human Papillomavirus in formalin-fixed paraffin-embedded tissue. Abbreviation: Immunohistochemistry (IHC); in situ hybridization (ISH); polymerase chain reaction (PCR); droplet-based digital PCR (ddPCR).

**Figure 3 viruses-13-01692-f003:**
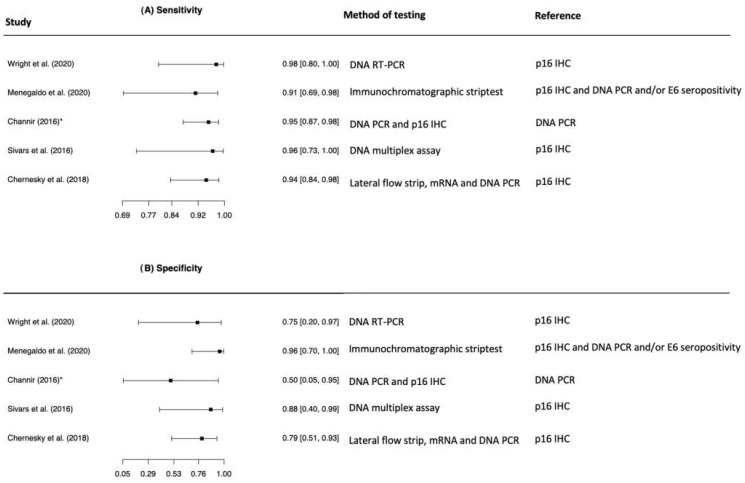
Forest plots of the meta-analysis estimating (**A**) Sensitivity with 95% confidence intervals, (**B**) Specificity with 95% confidence intervals in studies evaluating detection of Human Papillomavirus in fine needle aspiration.

**Figure 4 viruses-13-01692-f004:**
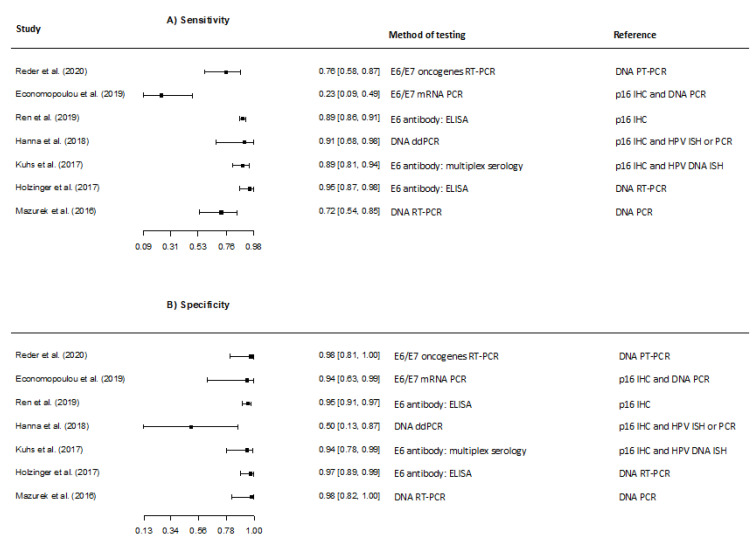
Forest plots of the meta-analysis estimating (**A**) Sensitivity with 95% confidence intervals, (**B**) Specificity with 95% confidence intervals in studies evaluating detection of Human Papillomavirus DNA in blood.

**Figure 5 viruses-13-01692-f005:**
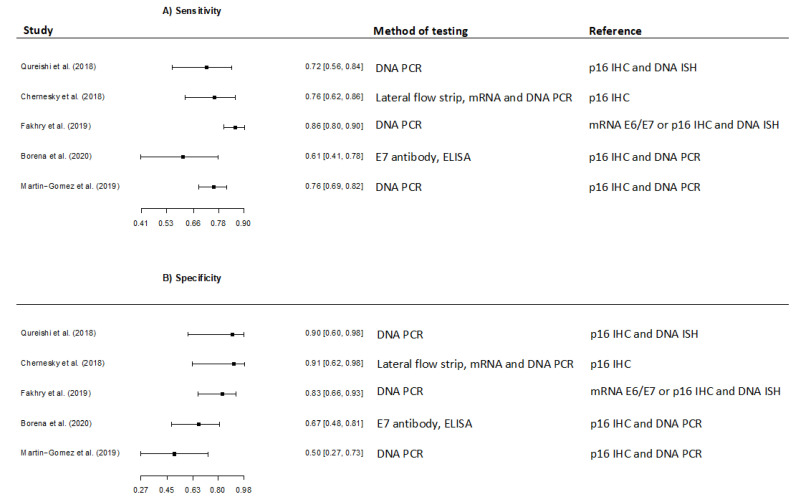
Forest plots of the meta-analysis estimating (**A**) Sensitivity with 95% confidence intervals, (**B**) Specificity with 95% confidence intervals in studies evaluating detection of Human Papillomavirus in oral samples.

**Table 1 viruses-13-01692-t001:** Overview of included studies.

Author (Year of Publication) Country [Study]	Study Period	OPSCC Patients. Total/HPV+	Sample	Method of Testing	Reference	True Positive	False Positive	False Negative	True Negative	Results
Henley-Smith et al. (2020) London [21]	2005–2016	100/38	FFPE	RNA ISH	p16 IHC, staining >70% and DNA ISH	38	18	0	44	NR. The 18 false positive cases were p16+/HPV-
Shengming et al. (2020) China [22]	2014–2019	257/47	FFPE	p16 IHC staining >70% DNA PCR	RNA ISH	p16: 44 DNA PCR: 47 p16/DNA PCR: 44	p16: 22 DNA PCR: 4 p16/DNA PCR: 4	p16: 3 DNA PCR: 0 p16/DNA PCR: 3	p16: 188 DNA PCR: 206 p16/DNA PCR: 206	p16 ISH: Sen: 93.6%; Spe: 89.10% DNA PCR: Sen: 97.9%; Spe: 97.6% p16 and DNA PCR: Sen: 91.7%; Spe: 98.1%
Mes et al. (2019) Netherlands [23]	2008–2011	80/20	FFPE	DNA RT-PCR	HPV16 E6 mRNA RT-PCR	19	0	1	55	NR
Randén-Brady et al. (2019) Finland [24]	Study I: 2000–2009 Study 2: 2012–2016	Study 1: 202/NR Study 2: 155/NR Study 1 + 2: HPV positive: 226	FFPE	Study 1: DNA ISH Study 2: DNA PCR Study 1 + 2: results combined: E6/E7 mRNA ISH	p16 IHC, staining >70%	mRNA: 211 DNA ISH: 101 DNA PCR: 91	mRNA: 10 DNA ISH: 4 HPV PCR: 5	mRNA: 15 DNA ISH: 16 HPV PCR: 18	mRNA: 121 DNA ISH: 81 HPV PCR: 41	mRNA ISH: Sen: 93.4%; Spe: 92.4% DNA ISH: Sen: 86.3%; Spe: 95.3% DNA PCR: Sen: 83.5%; Spe: 89.1%
Craig et al. (2019) Ireland [25]	2000–2011	221/90	FFPE	RNA-ISH DNA-ISH DNA-PCR	p16 IHC, staining >70%	DNA PCR: 71 DNA ISH: 67 RNA ISH:80	DNA PCR: 43 DNA ISH: 0 RNA ISH: 0	DNA PCR: 19 DNA ISH: 23 RNA ISH:10	DNA PCR: 88 DNA ISH: 131 RNA ISH: 131	DNA PCR: Sen: 79% (95% CI: 69–87); Spe: 67% (95% CI: 58–75) DNA ISH: Sen: 74% (95% CI: 64–83); Spe: 100% (95% CI: 87–100) RNA ISH: Sen: 88% (95% CI: 80–94); Spe: 100% (95% CI: 96–100)
Mena et al. (2018) Spain [26]	1990–2013	788/80	FFPE	DNA PCR p16 IHC, staining >70%	E6 mRNA RT-PCR	DNA PCR/p16: 58 p16: 58	DNA PCR/p16: 0 p16: 28	DNA PCR/p16: 9 p16: 9	DNA PCR/p16: 721 p16: 691	DNA PCR/p16: Sen: 86.6% (95% CI 76.0–93.7); Spe: 100.0% (95% CI 99.5–100.0). p16: Sen: 86.6% (95% CI 76.0–93.7); Spe: 96.1% (95% CI 94.4–97.4)
Meng et al. (2018) China [27]	2000–2016	1470/81	FFPE	DNA PCR	p16 IHC, staining >80%	78	0	3	1389	Sen: 100% Spe: 96%
Augustin et al. (2018) France [28]	2011–2013	126/56	FFPE	p16 IHC, staining >70% DNA-ISH RNA-ISH	DNA PCR	p16: 46 DNA ISH: 41 RNA ISH:49	p16: 7 DNA ISH: 2 RNA ISH: 2	p16: 10 DNA ISH: 15 RNA ISH: 7	p16: 68 DNA ISH: 63 RNA ISH: 68	p16: Sen: 82% (95% CI 70–91); Spe: 90% (95% CI 80–96) DNA ISH: Sen: 73% (95% CI 60–84); Spe: 97% (95% CI 90–100) RNA ISH: Sen: 88% (95% CI 76–95); Spe: 97% (95% CI 90–100) p16 and DNA ISH: Sen: 88% (95% CI 76–95); Spe: 97% (95% CI 90–100) p16 and RNA ISH: Sen: 95% (95% CI 85–99); Spe: 100% (95% CI 92–100)
Mendez-Pena et al. (2017) Boston, USA [29]	2015–2016	57/26	FFPE	RNA ISH DNA ISH p16 IHC, staining >50%	DNA PCR	RNA ISH: 26 DNA ISH: 23 p16: 24	RNA ISH: 4 DNA ISH: 8 p16: 13	RNA ISH: 0 DNA ISH: 3 p16: 1	RNA ISH: 27 DNA ISH: 23 p16: 16	RNA ISH: Sen: 100%; Spe: 87% DNA ISH: Sen: 88%; Spe: 74% p16: Sen: 96%; Spe: 55%
Wright et al. (2020) Tennessee, USA [30]	NR	20/19	FNA	DNA RT-PCR	p16 IHC, staining >70%	19	0	0	1	Sen: 100%; Spe: 100%
Menegaldo et al. (2020) Italy [31]	2016–2019	29/16	FNA	HPV16 and HPV18 E6 oncoproteins, lateral flow immunochromatographic strip test	p16 IHC, staining >70% combined with DNA PCR and/or E6 seropositivity	15	0	1	11	Sen: 94% (95% CI: 70–100); Spe: 100% (95% CI: 72–100)
Channir (2016) Denmark [32]	2002–2016	71/71 (HPV- group: 47/7 with OSCC, 20 Warthin’s tumour, 20 branchial cleft cyst)	FNA	DNA PCR p16 IHC, staining <75%	DNA PCR	68	0	3	47	Sen: NR. Spe: 100% (95% CI 92.5–100.0)
Sivars et al. (2016) Sweden [33]	2013–2016	16/13	FNA	DNA multiplex assay	p16 IHC, staining >70%	13	0	0	3	Sen: 100%; Spe: 100%
Chernesky et al. (2018) Canada [34]	NR	59/48	Saliva and oral swabs (BOT and tonsillar area) pooled and FNA	(1) OncoE6 proteins–lateral flow strip (2) HPV E6/E7 mRNA assay (3) DNA PCR	p16 IHC, staining >70%	Oral sample: (1) 3 (2) 22 (3) 35 FNA: (1) 38 (2) 46 (3) 42	Oral sample: (1) 0 (2) 0 (3) 1 FNA: (1) 1 (2) 1 (3) 2	Oral sample: (1) 45 (2) 26 (3) 11 FNA: (1) 10 (2) 2 (3) 2	Oral sample: (1) 11 (2) 11 (3) 10 FNA: (1) 10 (2) 10 (3) 9	NR
Borena et al. (2020) Austria [35]	2018–2020	50/23	Cytology brush tests of tumour surface	E7 antigen test, ELISA	p16 IHC, staining >66% and DNA PCR	14	9	9	18	Sen: 60.9% (95% CI 38.5–80.3); Spe: 66.7% (95% CI 46–83.5)
Martin-Gomez et al. (2019) Florida, USA [36]	2014–2017	171/157	Oral rinse	DNA PCR	p16 IHC, staining >70% and DNA PCR	119	7	38	7	Sen: 75.8%; Spe: 50.0%
Fakhry et al. (2019) Ohio and Baltimore USA [37]	2011–2016	217/187	Oral rinse	DNA PCR	mRNA E6 or E7 or p16 IHC/combined with DNA ISH	161	5	26	25	NR
Qureishi et al. (2018) United Kingdom [38]	2015–2016	46/36	Oral rinse	DNA PCR	p16 IHC, staining >70% and DNA ISH. Positive if: p16+/no HPV DNA test, p16+/HPV DNA+ or p16- and HPV DNA+.	p16/HPV: 26	p16/HPV: 1	p16/HPV: 10	p16/HPV: 9	Oral rinse vs. p16: Sen: 73.5% (95% CI 55.6–87.1); Spe: 83.3% (95% CI 51.6–97.9). Oral rinse vs. DNA ISH: Sen: 66.7% (95% CI 43–85.4); Spe: 87.5% (95% CI 47.4–99.7). Oral rinse vs. p16/HPV: Sen: 72.2 (95% CI 54.8–85.8); Spe: 90 (95% CI 55.5–99.8)
Reder et al. (2020) Germany [39]	2014–2017	48/28	Plasma	E6 and E7 oncogenes RT-PCR	HPV16-DNA RT-PCR	23	0	7	20	Sen: 77%; Spe: 100%
Economopoulou et al. (2019) Ohio, USA [40]	NR	22/14	Blood (CtC)	HPV16 E6/E7 mRNA qPCR	p16 IHC, staining > 70% and HPV DNA qPCR	3	0	11 (7 HPV16)	8	NR
Chera et al. (2019) North Carolina, USA [41]	2016 –2018	103/103. 155 controls (55 healty controls and 60 non-HPV malignancies (not OPSCC))	Plasma	DNA, ddPCR	p16 IHC, staining > 70%	84	Control: 3	19	Control: 112	Sen: 89%; Spe: 97%
Ren et al. (2019) China [42]	2007–2017	783/611	Plasma	HPV16 E6 antibody advanced multiplex analysis/ELISA	p16 IHC, staining >70%	545	8	66	164	Sen: 89% (95%CI 86–92); Spe: 95% (95%CI 91–98)
Damerla et al. New York, USA (2019) [43]	NR	97/97. (HPV- group: 7 HPV- HNC and 20 healthy controls)	Plasma	HPV16 and HPV33 ddPCR	p16 IHC, staining >70% or DNA ISH, or RNA ISH	90	0	7	27	Sen: 92.8%; Spe: 100%
Hanna et al. (2018) Boston, USA [44]	2017–2018	17/15	Blood (Obtained at any time during treatment)	DNA ddPCR	p16 IHC, staining >70% and DNA ISH or PCR	14	1	1	1	Sen: 93.3% (95%CI 68.0–99.8); Spe: 50% (95%CI 1.3–98.7)
Kuhs et al. (2017) Pittsburgh, USA [45]	2003–2013	112/87	Serum	HPV16 E6 multiplex serology	p16 IHC, staining >70% and DNA ISH	78	1	9	24	Sen: 89.7% (95%CI, 81.3–95.2); Spe: 96.0% (95% CI, 79.6–99.9)
Holzinger et al. (2017) Germany and Italy [46]	NR	120/66	Serum	HPV16 E6 antibody ELISA	HPV16 DNA, RT-PCR	63	1	3	53	Sen: 96% (95%CI 88–98); Spe: 98% (95%CI 90–100)
Mazurek et al. (2016) Poland [47]	2011–2013	51/29	Plasma	HPV16 DNA, RT-PCR	HPV16 DNA qPCR	21	0	8	22	Sen: 72%; Spe: 100%.

Abbreviation: Oropharyngeal squamous cell carcinoma (OPSCC); Human Papillomavirus (HPV); 95% confidence intervals (95% CI); Immunohistochemistry (IHC); In situ hybridization (ISH); Droplet-based digital PCR (ddPCR); Reverse transcriptase polymerase chain reaction (RT-PCR); Not reported (NR); Sensitivity (sen); Specificity (SPE); Unstimulated whole mouth saliva (UWMS); Formalin-fixed paraffin-embedded (FFPE); Fine needle aspiration (FNA); Base of tongue (BOT); Circulating tumour cells (CtC).

**Table 2 viruses-13-01692-t002:** Pooled sensitivity and specificity for different Human Papillomavirus (HPV) detection methods in formalin-fixed paraffin-embedded oropharyngeal cancer tissue.

	Number of Studies Included in Meta-Analysis	References:	Sensitivity (95% CI)	Specificity (95% CI)
RNA ISH	5	p16 IHC: 3 studies DNA PCR: 2 studies	93.1 (87.4–96.4)	91.9 (78.8–97.2)
DNA ISH	4	p16 IHC: 2 studies DNA PCR: 3 studies	81.1 (71.9–87.8)	94.9 (79.1–98.9)
DNA PCR	5	p16 IHC: 3 studies RNA: 2 studies	90.4 (81.4–95.3)	81.1 (71.9–87.8)
p16 IHC	4	RNA: 2 studies DNA PCR: 2 studies	83.3 (69.0–91.8)	93.5 (88.4–96.5)

Abbreviation: Immunohistochemistry (IHC); In situ hybridization (ISH); Polymerase chain reaction (PCR); 95% confidence intervals (95% CI).

## Data Availability

Not applicable.

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
