# Peer review of "Diagnostic Accuracy of HPV Detection in Patients with Oropharyngeal Squamous Cell Carcinomas: A Systematic Review and Meta-Analysis"

_viruses, 2021, doi:10.3390/v13091692_

Round 1

Reviewer 1 Report

This review by Katherine Kronberg Jakobsen, et al. evaluated the literature reporting the diagnostic accuracy of HPV diagnostic techniques in OPSCC patients. They performed meta-analysis on 27 studies reporting a variety of HPV detection methods for different types of samples (FFPE, FNS, blood and oral). The pooled sensitivity and specificity were estimated for different detection methods in different samples using p16 IHC and/or DNA PCR as the reference method.  The results showed an overall high accuracy of HPV detection methods in tumor tissues regardless of the detection methods and the HPV detection in blood samples may provide a promising new way of HPV detection.

The summary of sensitivity/specificity of different HPV detection methods in OPSCC patients if of particular interest in this field. However, there are some results/methods need to be clarified.

  1. The reference methods varied for different studies. How the meta-analysis took that into account? Are the sensitivities/specificities comparable if they were calculated using different standards.
  2. The meta-analysis method is not clear. Please elaborate the methods. 
  3. The reference methods were not clear in the forest plots of sensitivity/specificity in FNA, blood and oral samples (Figure 3-5). Please add the annotation as in Figure 2.
  4. A typo in line 211 (“nooted”).

Author Response

Reviewer 1:

This review by Katherine Kronberg Jakobsen, et al. evaluated the literature reporting the diagnostic accuracy of HPV diagnostic techniques in OPSCC patients. They performed meta-analysis on 27 studies reporting a variety of HPV detection methods for different types of samples (FFPE, FNS, blood and oral). The pooled sensitivity and specificity were estimated for different detection methods in different samples using p16 IHC and/or DNA PCR as the reference method.  The results showed an overall high accuracy of HPV detection methods in tumor tissues regardless of the detection methods and the HPV detection in blood samples may provide a promising new way of HPV detection.

The summary of sensitivity/specificity of different HPV detection methods in OPSCC patients if of particular interest in this field. However, there are some results/methods need to be clarified.

Thank you for your helpful comments and for considering our manuscript. We have answered the all comments point by point below.

  1. The reference methods varied for different studies. How the meta-analysis took that into account? Are the sensitivities/specificities comparable if they were calculated using different standards.

Reply: This is an important point. To highlight and elaborate on this we included a section in the discussion regarding the limitation of including studies with different reference methods and a discussion of the challenges this might result in.

  1. The meta-analysis method is not clear. Please elaborate the methods. 

Reply: We have elaborated on the method of the meta-analysis. We hope that the method of the meta-analysis is clearer and more understandable now.

  1. The reference methods were not clear in the forest plots of sensitivity/specificity in FNA, blood and oral samples (Figure 3-5). Please add the annotation as in Figure 2.

Reply: We have added the reference method and detection method to all of the figures to enhance clarification.

  1. A typo in line 211 (“nooted”).

Reply: Thank you for pointing this out. This has been corrected.

Reviewer 2 Report

The aim of this study was to review the manuscripts retrieve from the period of the last five years on the methods of HPV detection and to assess the diagnostic accuracy for HPV detection in patients with head and neck cancer based on detection methods and in different sample types. Human papillomavirus (HPV)-associated head and neck squamous cell carcinomas (HNSCC) represent an increasing health problem, particularly in the oropharyngeal tonsils.

The main disadvantage of this meta-analysis is the limitation of article searches to the last 5 years only. However, the methods that the article evaluates are relatively old and have been used for a long time. Our knowledge of the role of HPV in head and neck cancer is also relatively old. The first articles began to appear in the early 90's of the last century. Our knowledge of the role of HPV in head and neck cancer is also relatively old. The first articles began to appear in the early 90's of the last century. Simultaneously with the discovery of HPV in head and neck tumours, the development of the most sensitive and specific diagnostic approaches began. The main articles that dealt with diagnostics very intensively were published before 2016 and are therefore not included in this meta-analysis.

At the beginning of this study, a large number of articles were identified, of which only 27 were selected for further analysis. It is not clear why most of the articles were excluded. These 27 studies detect HPV in different materials, but they need different approaches to detection.

A major drawback of the study is the fact that the 27 articles use different methods of the "gold standard of HPV detection, some use HPV RNA detection, some only DNA and some combine DNA and p16 immunohistochemistry. E6/E7 HPV-mRNA evaluation is considered the golden standard to assess HPV positivity.

It is not very clear what the authors wanted to share with this meta-analysis, what new this study brings.

Author Response

Reviewer 2:

The aim of this study was to review the manuscripts retrieve from the period of the last five years on the methods of HPV detection and to assess the diagnostic accuracy for HPV detection in patients with head and neck cancer based on detection methods and in different sample types. Human papillomavirus (HPV)-associated head and neck squamous cell carcinomas (HNSCC) represent an increasing health problem, particularly in the oropharyngeal tonsils.

Thank you for your helpful comments and for reviewing our manuscript. We have answered the all comments point by point below.

The main disadvantage of this meta-analysis is the limitation of article searches to the last 5 years only. However, the methods that the article evaluates are relatively old and have been used for a long time. Our knowledge of the role of HPV in head and neck cancer is also relatively old. The first articles began to appear in the early 90's of the last century. Our knowledge of the role of HPV in head and neck cancer is also relatively old. The first articles began to appear in the early 90's of the last century. Simultaneously with the discovery of HPV in head and neck tumours, the development of the most sensitive and specific diagnostic approaches began. The main articles that dealt with diagnostics very intensively were published before 2016 and are therefore not included in this meta-analysis.

Reply: Thank you for your important feedback and for sharing your knowledge regarding the HPV detection methods as you describe above. It is true that it is a limitation to the study that only articles published within the last five years were included. We only included articles published within the last five years as the guidelines from the American Society of Oncology for defining a tumour as p16+ by a cut-off of 70% nuclear and cytoplasmic staining were renewed and published in the last five years (Lewis, J.S.; Beadle, B.; Bishop, J.A.; Chernock, R.D.; Colasacco, C.; Lacchetti, C.; Moncur, J.T.; Rocco, J.W.; Schwartz, M.R.; Seethala, R.R.; et al. Human papillomavirus testing in head and neck carcinomas guideline from the college of American pathologists. Arch. Pathol. Lab. Med. 2018, 142, 559–597, doi:10.5858/arpa.2017-0286-CP.), and as p16 is both a very used reference method and a detection method we included recent studies to avoid excessive variability in the included studies. A previous review regarding the correlation between HPV and p16 in OPSCC patients found that a majority of studies used a minimum of 5-69% staining oppose the recent guidelines from the American Society of Oncology. We have added a section regarding this and highlighting the limitation this comprises to the discussion. We hope that you will find this fulfilling.

At the beginning of this study, a large number of articles were identified, of which only 27 were selected for further analysis. It is not clear why most of the articles were excluded.

Reply: Thank you for pointing this out. We have added a section to the result section elaborating on the exclusion process. Furthermore, the reasons for the exclusion of the articles is depicted in Figure 1.

These 27 studies detect HPV in different materials, but they need different approaches to detection. A major drawback of the study is the fact that the 27 articles use different methods of the "gold standard of HPV detection, some use HPV RNA detection, some only DNA and some combine DNA and p16 immunohistochemistry. E6/E7 HPV-mRNA evaluation is considered the golden standard to assess HPV positivity.

Reply: This is a very good and valid point. We have included a section in the discussion regarding the limitation of including studies with different reference methods and a discussion of the challenges this might result in.

It is not very clear what the authors wanted to share with this meta-analysis, what new this study brings.

Reply: This is an important point. To highlight the importance of this review we have added a section to the introduction highlighting the importance of this review and explained that the review is a comparison of diagnostic accuracy in different specimens which is warranted. Furthermore, we have clarified the conclusion to highlight this matter.

Reviewer 3 Report

The Systematic review entitled “Diagnostic accuracy of HPV detection in patients with Oropha-2 ryngeal Squamous Cell Carcinomas: A systematic review and 3 meta-analysis” reports the evaluation of literature reporting the diagnostic accuracy of Human Papillomaviruses (HPV) diagnostic techniques in patients with oro-pharyngeal cancer.

The authors reported the thirteen studies evaluated HPV detection in tumor tissues and nine studies examined HPV detection in blood samples as well as five studies for HPV detection in oral samples.

The manuscript consists of the comparison of the evaluation methods of HPV detection, remarking the HPV blood samples detection as promising way of HPV detection. Although the aim of this review is of interest, there are some questions to be addressed and the manuscript should be  revised. The Introduction section is well stated and well organized. Although the research on head and neck tumors is very limited, the data collected are sufficient to carry out the required analyzes.

Reviewer report:

Major revision

  1. The discussion section should be enriched and implemented with the limits of the methods analyzed.

Minor revision

  1. Please align the percentages with the latest WHO/IARC data.
  2. The table 1 is very informative but it is difficult to read due to the lack of dividing lines between the various columns. Please rework the table 1 to improve the readability.
  3. Page 10 line 171. Please add dot at the end of the sentence.
  4. Page 13 lane 221. Please correct the verb: “it should be noted”
  5. Page 13 lane 235. Please correct the verb: “excists”
  6. Page 13 lane 235. Please correct the verb: “indicatest”
  7. Page 13 lane 252. Please correct the word: “limitaion”
  8. The manuscript would benefit from a careful read for English editing to improve the readability.

Author Response

 Reviewer 3:

The Systematic review entitled “Diagnostic accuracy of HPV detection in patients with Oropha-2 ryngeal Squamous Cell Carcinomas: A systematic review and 3 meta-analysis” reports the evaluation of literature reporting the diagnostic accuracy of Human Papillomaviruses (HPV) diagnostic techniques in patients with oro-pharyngeal cancer.

The authors reported the thirteen studies evaluated HPV detection in tumor tissues and nine studies examined HPV detection in blood samples as well as five studies for HPV detection in oral samples.

The manuscript consists of the comparison of the evaluation methods of HPV detection, remarking the HPV blood samples detection as promising way of HPV detection. Although the aim of this review is of interest, there are some questions to be addressed and the manuscript should be  revised. The Introduction section is well stated and well organized. Although the research on head and neck tumors is very limited, the data collected are sufficient to carry out the required analyzes. 

Thank you for your constructive feedback and comments. For clarification we have answered the all comments point by point below.

Reviewer report:

Major revision

  1. The discussion section should be enriched and implemented with the limits of the methods analyzed.

Reply: This is a good point. We have included a section in the discussion regarding the limitation of the analysis.

Minor revision

  1. Please align the percentages with the latest WHO/IARC data.

Reply: Thank you for the feedback. We have added two sections to the discussion regarding the latest data from the IARC and NCI and compared their data to our results. This has put our data into perspective and thus hopefully improve the article.

  1. The table 1 is very informative but it is difficult to read due to the lack of dividing lines between the various columns. Please rework the table 1 to improve the readability.

Reply: We have added dividing lines to enhance readability and revised the table. We hope this will improve readability.

  1. Page 10 line 171. Please add dot at the end of the sentence.

Reply: Thank you for pointing this out. This has been added.

  1. Page 13 lane 221. Please correct the verb: “it should be noted”

Reply: This has been corrected. Again, thank you for pointing it out.

  1. Page 13 lane 235. Please correct the verb: “excists”

Reply: This has been corrected.

  1. Page 13 lane 235. Please correct the verb: “indicatest”

Reply: Thank you again. This has been corrected.

  1. Page 13 lane 252. Please correct the word: “limitaion”

Reply: This has been corrected.

  1. The manuscript would benefit from a careful read for English editing to improve the readability.

Reply: This is a very valid point. The manuscript has undergone a careful read for English editing to improve the manuscript.

Round 2

Reviewer 2 Report

The authors modified the manuscript, explained all the comments, and discussed the problems of the article in the discussion section. However, these problems remain, the variability of methods and the use of the gold standard for HPV detection in individual articles is very high. I'm also a little worried that some of the articles may have dropped out due to the great selection, although they would be very beneficial

Reviewer 3 Report

The authors revised the draft by providing additional useful information and improving the readability. So the reviewer approves the manuscript publication.

Note: the caption of figure 2 is not readable in the pdf file of the draft.